# Exploring the Hemostatic Effects of Platelet Lysate-Derived Vesicles: Insights from Mouse Models

**DOI:** 10.3390/ijms25021188

**Published:** 2024-01-18

**Authors:** Nobuhisa Hirayu, Osamu Takasu

**Affiliations:** Department of Emergency and Critical Care Medicine, Kurume University School of Medicine, 67 Asahi-machi, Kurume 830-0011, Japan; takasu_osamu@kurume-u.ac.jp

**Keywords:** hemostasis, platelet-derived extracellular vesicles, platelet lysate

## Abstract

Platelet transfusion has various challenges, and platelet-derived extracellular vesicles have been reported to have more significant procoagulant activity than platelets themselves. Furthermore, platelet products derived from platelet-rich plasma and platelet lysates (PLs) have gained attention for their physiological activity and potential role as drug delivery vehicles owing to the properties of their membranes. We aimed to investigate the characteristics of the fractions isolated through ultracentrifugation from mouse-washed PLs and assess the potential clinical applications of these fractions as a therapeutic approach for bleeding conditions. We prepared PLs from C57BL/6 mouse-washed platelets and isolated three different fractions (20K-vesicles, 100K-vesicles, and PLwo-vesicles) using ultracentrifugation. There was a notable difference in particle size distribution between 20K-vesicles and 100K-vesicles, particularly in terms of the most frequent diameter. The 20K-vesicles exhibited procoagulant activity with concentration dependence, whereas PLwo-vesicles exhibited anticoagulant activity. PLwo-vesicles did not exhibit thrombin generation capacity, and the addition of PLwo-vesicles to Microparticle Free Plasma extended the time to initiate thrombin generation by 20K-vesicles and decreased the peak thrombin value. In a tail-snip bleeding assay, pre-administration of 20K-vesicles significantly shortened bleeding time. PL-derived 20K-vesicles exhibited highly potent procoagulant activity, making them potential alternatives to platelet transfusion.

## 1. Introduction

Deaths from traffic injuries are a major global concern, and resultant hemorrhage is an important cause of acute-phase mortality [1]. Platelet transfusion, with its hemostatic action, is an important means to control acute bleeding [2] and is associated with various risks, such as allergic reactions, transfusion-related acute lung injury, and thrombosis [3,4,5]. Additionally, platelet transfusion products have a short shelf life (approximately 5 days in many countries) and the activation of platelets during storage is also a concern [6].

Hemostasis is an important physiological process that prevents bleeding after vessel injury and involves two main mechanisms—blood coagulation and platelet activation [7]. Platelets are recruited to a site of vessel injury, and the activated platelets release several proteins and molecules from α-granules and dense granules or as platelet-derived extracellular vesicles (PD-EVs), which promote continued platelet aggregation, coagulation, inflammation, and vasoconstriction. Interestingly, the surface of a type of PD-EV that is released with platelet activation has been reported to possess significantly higher procoagulant activity, approximately 50–100 times more potent than the surface of an activated platelet [8]. These findings suggest that PD-EVs can be used as therapeutic agents for hemostatic strategies.

Platelets have various functions and characteristics beyond their hemostatic roles. Platelets themselves are particularly rich in growth and angiogenic factors; therefore, platelet products, such as platelet-rich plasma (PRP) and platelet lysate (PL), which are prepared ex vivo, are useful in wound healing and regenerative medicine [9,10]. Furthermore, in recent years, PD-EVs or platelet-derived vesicles have gained attention not only because of their physiological activity but also because of their potential role as drug delivery vehicles owing to the properties of their membranes [11,12,13].

Platelet-derived products or platelet-derived vesicles can be produced ex vivo from platelet transfusion products or artificially activated platelets through various means. PL is a platelet product that allows the reproducibility of platelet content extraction without the use of chemical substances. PL contains abundant vesicles [14], and these PL-derived vesicles have been investigated for clinical applications [15] under various conditions and pathologies, including fracture treatment and cartilage regeneration [16,17], wound healing in skin ulcers [18], neurodegenerative diseases and trauma, and COVID-19-related lung disorders [19]. There is also a suggestion of clinical applications for PL-derived vesicles in hemorrhagic shock [20], although research in this area remains limited [21]. Moreover, it is unclear whether the PL-derived vesicles that did not have physiologically activated processes had similar procoagulant activity as that of PD-EVs released from activated platelets in vivo [22]. We, therefore, aimed to investigate the characteristics of the fractions isolated through ultracentrifugation from mouse PLs and assess the potential clinical applications of these fractions as a therapeutic approach for bleeding conditions.

## 2. Results

### 2.1. Particle Size Distribution and Concentration of Mouse PL (mPL)-Derived Vesicles

The typical particle size distributions of both mPL-derived vesicles obtained through centrifugation at 20,000× *g* (20K-vesicles) and mPL-derived vesicles obtained by collecting the supernatant obtained after centrifugation at 20,000× *g* and further centrifugation at 100,000× *g* (100K-vesicles) are shown in Figure 1. The mean most frequent diameters of 20K-vesicles and 100K-vesicles were 130 nm and 96 nm, respectively, and the particle concentrations were 1.84 × 10^12^ particles/mL for 20K-vesicles and 3.83 × 10^11^ particles/mL for 100K-vesicles. There were clear differences in the distributions of the 20K-vesicles and 100K-vesicles.

Following 20,000× *g* centrifugation, the mPL was centrifuged at 100,000× *g*. The most frequent diameters for the 20K-vesicles and 100K-vesicles were 144.4 nm and 87.3 nm, respectively, with clear differences between the two. Washed platelet samples were prepared using three units (U) per sample. Regarding the platelet-derived vesicles, the particle concentrations were measured three times in each sample, and the mean and mode values were calculated. As shown in Appendix A, we observed some between-group differences in the number of particles of the 20K-vesicles and/or 100K-vesicles in each sample; however, we considered that these did not have any major impact on the results.

### 2.2. Comparison of Thrombin Generation Capacity by Three mPL-Derived Fractions

Figure 2a and Appendix A show the typical thrombin generation curves for three mPL-derived fractions. The 20K-vesicles exhibited a distinct and concentration-dependent thrombin generation capacity (Figure 2b). In some samples, a slight thrombin generation capacity was observed with 100K-vesicles; however, none of the samples showed thrombin generation capacity with the PL without the vesicle fraction (the supernatant after ultracentrifugation at 100,000× *g* was designated as the PLwo-vesicles).

In contrast, in samples in which the PLwo-vesicle fraction was added to microparticle-free plasma (MpFP), the first burst in thrombin formation (lag phase time) upon the addition of the 20K-vesicles was prolonged, and a decrease in the peak thrombin value was observed (Figure 3).

### 2.3. Platelet Intracellular Proteins in the PLwo-Vesicle Fraction

The mean Angiopoietin-1 levels in the PLwo-vesicles were significantly higher in the PLwo-vesicle fraction at 159.9 ± 11.5 ng/mL compared to 1.27 ± 0.84 ng/mL in the MpFP. Similarly, Dickkopf-1 (DKK 1) was higher at 2462 ± 361 pg/mL in the PLwo-vesicles compared to 73 ± 48 pg/mL in MpFP; the hepatocyte growth factor (HGF) was elevated at 6158 ± 105 pg/mL in the PLwo-vesicles compared to 103.4 ± 57.8 pg/mL in the MpFP, and serpinE2/protease nexin-1(SERPINE2) was significantly higher at 79.2 ± 5.9 µg/mL in the PLwo-vesicles compared to 0.3 ± 0.1 µg/mL in the MpFP, thus barely detectable in the MpFP. In contrast, the tissue factor pathway inhibitor (TFPI) showed markedly higher levels in the MpFP at 275.0 ± 16.6 ng/mL compared to 10.1 ng/mL ± 1.7 in the PLwo-vesicles. In the ELISA, three samples were measured for each fraction.

### 2.4. Phosphatidylserine and Factor X in 20K-Vesicles

A flow cytometry analysis confirmed Annexin V expression in 20K-vesicles, as shown in Figure 4. In addition, 15.0 ± 0.3 ng/mL of Factor X was detected using the ELISA analysis.

### 2.5. Procoagulant/Anticoagulant Effects of the Three mPL-Derived Fractions In Vitro

The rotational thrombo-elastometry (ROTEM Delta, Werfen, Barcelona, Spain) measurement results are presented in Figure 5 and Figure 6 and Appendix A. Figure 5 shows a typical coagulation process in ROTEM when each of the three mPL-derived fractions was added into diluted whole blood with PBS as a coagulation trigger. The measurement results for the following parameters are shown in Figure 6: CT (Clotting Time), which represents the time from the start of the test to the initial fibrin formation; CFT (Clot Formation Time), which is a dynamic parameter indicating the speed of clot formation through fibrin polymerization; and the maximum clot firmness (MCF) index, which reflects the firmness of the clot. In a comparison based on the control group (PBS), CT significantly reduced only when 20K-vesicles were added. Contrarily, a simple comparison of measurements between the four groups showed significant differences between the addition of 20K-vesicles and PLwo-vesicles for CT, CFT, and MCF (Appendix A).

### 2.6. Hemostatic Effect of 20K-Vesicles in a Bleeding Model (Tail-Snip Bleeding Assay)

The median time required for hemostasis when 20K-vesicles or PBS (control) were intraperitoneally administered 60 min before tail amputation was 80 s for 20K-vesicles compared to 394 s for PBS, showing a significant reduction in hemostasis time with 20K-vesicle administration (Figure 7). There was no significant difference in the absorbance of blood clots, which reflected the bleeding volume, between the groups.

### 2.7. Thrombin–Antithrombin Complex (TAT) and Microthrombus in Various Organs 24 h after the Administration of 20K-Vesicles

At 24 h after the tail-snip bleeding assay, plasma TAT levels were compared between the groups. TAT levels in the 20K-vesicle-administration group were 19.7 ± 2.5 pg/mL, and those in the PBS-administration group were 21.9 ± 2.5 pg/mL, with no significant difference observed between the groups (*p* = 0.22, *n* = 5 in each group). No apparent thrombus formation was observed in the liver, kidneys, or lungs (Appendix A).

### 2.8. Cytokine Concentration Associated with 20K-Vesicle Administration

After administration of 0.5 U of 20K-vesicles or PBS as controls, the plasma levels of TNF-alpha, IL-4, and IL-6 were evaluated 6 h later. However, the levels of these cytokines were below the measurement sensitivity in both groups (*n* = 5 in each group).

## 3. Discussion

There was a notable difference in particle size distribution between the 20K-vesicles and 100K-vesicles, particularly in terms of the most frequent diameter. The 20K-vesicle fraction derived from mPL exhibited strong procoagulant activity, whereas the PLwo-vesicle fraction exhibited anticoagulant activity.

Although the 20K-vesicles in this study are vesicles that have not undergone an in vivo physiological activation, significant procoagulant activity was observed in them. Moreover, this procoagulant activity was also similar to the platelet-transfusion-product-derived vesicles [23,24,25]. In the 20K-vesicles, the expression of phosphatidylserine (PS) and coagulation Factor X (Factor X) was confirmed. PS, which is expressed on the surface of activated platelets, is considered a central regulatory molecule in blood coagulation [26] and, during activation, the PS present on the inner side of the platelet membrane is externalized to the outer membrane. Furthermore, PS is observed in vesicles released from platelets, and the extent of its expression on the vesicles is reported to be higher than that on the surface of activated platelets, exhibiting 50–100 times greater procoagulant activity [8]. Additionally, it is believed that Factor X is either released from platelets or transferred to the vesicle surface [27] and is considered one of the key factors contributing to the significant procoagulant activity observed in 20K-vesicles.

In contrast, 100K-vesicles showed only slight thrombin-generating capacity, which was observed only when the administered dose was increased. We believe that this represents the limitation of separating physiological activity of the vesicles through ultracentrifugation. However, from the perspective of isolating vesicles with procoagulant activity, the separation achieved at 20,000× *g* (20K-vesicles) was considered sufficient.

In contrast, the PLwo-vesicle fraction showed no thrombin-generating capacity in the thrombin generation assay (TGA), and thrombo-elastometry revealed a clear anticoagulant activity opposite to that of 20K-vesicles. To assess the impact of PLwo-vesicles on the procoagulant activity of 20K-vesicles, we conducted TGA, which demonstrated that the addition of PLwo-vesicles reduced thrombin generation by 20K-vesicles and extended the time to thrombin generation. This indicated that PLwo-vesicles have an inhibitory effect on the procoagulant activity of 20K-vesicles.

Based on these results, we examined two proteins with reported anticoagulant activities that are known to be present in platelets: TFPI and serpinE2/protease nexin-1. TFPI is an anticoagulant protein that inhibits tissue factor VIIa (TF-FVIIa) and Factor Xa (FXa) and has two isoforms, TFPIα and TFPIβ. TFPIα is released from vascular endothelial cells, megakaryocytes, and platelets. The localization within platelets is reported to be associated with components other than α granules and lysosomes [28]. It is released upon platelet activation and plays a role in limiting the growth of thrombi following vascular injury, thereby contributing to the regulation of thrombus formation [29]. In contrast, serpinE2/protease nexin-1 is the most effective and potent serine protease inhibitor against thrombin, plasminogen activators, and plasmin [30]. It is minimally present in plasma and is found in various cells, including platelets and vascular endothelial cells. In platelets, it is present on the platelet surface and within α granules, where it serves as a negative regulator (having anticoagulant and antithrombotic properties) during the process of thrombus generation [31].

Considering the isolation process used in this study, we hypothesized that various physiologically active substances present in platelet granules would be found at high concentrations in PLwo-vesicles. Therefore, the concentrations of TFPI and serpinE2/protease nexin-1 in PLwo-vesicles, along with representative proteins known to be present in ɑ granules, such as Angiopoietin-1, HGF, and DKK 1, which can be stably measured using commercially available kits, were examined, with MpFP serving as a control.

SerpinE2/protease nexin-1 was present at significantly higher concentrations in the PLwo-vesicle fraction, along with Angiopoietin-1, HGF, and DKK 1, while it was barely detectable in MpFP. In contrast, TFPI was detected in both MpFP and PLwo-vesicle fractions, which is consistent with previous reports [32]. Because it was present in higher concentrations in the plasma in this experiment, it was suggested that the anticoagulant activity of the PLwo-vesicle fraction was primarily attributed to the action of serpinE2/protease nexin-1. For the clinical application of platelet-derived vesicles in hemorrhagic conditions, the 20K-vesicle fraction is expected to be extracted and adjusted as a fraction with more potent procoagulant activity by excluding the PLwo-vesicle fraction, which has at least anticoagulant activity from the PL.

In the tail-snip bleeding assay, administration of 20K-vesicles significantly shortened the bleeding time. In the 20K-vesicle group, the sustained oozing bleeding time, which was observed in the late phase of bleeding in the PBS-treatment group, was significantly shorter. We assume that this may be the reason for the shortened bleeding time. Contrarily, a previous report [33] that examined the function of platelet-derived vesicles collected from stored human platelet blood in a similar tail-snip bleeding assay reported that the administration of platelet-derived vesicles reduced the amount of bleeding. In this study, a reduction in blood loss could not be demonstrated. We amputated the tail more proximally, at a length of 15 mm, compared with previous studies [33]. This resulted in the resection of a larger-diameter blood vessel, leading to a significant amount of bleeding immediately after vascular resection (early phase of bleeding), which then strongly influenced the total bleeding. We considered this as the reason why no significant difference was observed in the amount of total bleeding between the two groups. Further research is required to determine whether the difference in blood loss compared with those in previous studies is due to differences in the coagulant activity of platelet-derived vesicles.

When considering the systemic administration of 20K-vesicles to promote hemostasis and coagulation, there are concerns regarding potential side effects, such as thrombus formation and extension to deep vein thrombosis due to potent procoagulant (hypercoagulability) activity [34]. In this study, mice were evaluated for bleeding time and volume, and no deterioration in health or mortality was observed within 24 h after administration. Furthermore, there was no significant increase in TAT levels 24 h after 20K-vesicle administration, and there was no evidence of sustained thrombin generation. Additionally, no promotion of thrombus generation was observed, and there was no indication of increased thrombus formation in the liver, kidney, or lung tissues 24 h after administration compared to the control group receiving PBS.

Moreover, it has been suggested that EVs released from various cells interact with the vascular endothelium and immune cells, affecting various immune cells such as granulocytes, lymphocytes, and monocytes [35]; however, it is unclear whether EVs contribute to the silent clearance of EVs [36,37], modulate some form of inflammatory response, or influence inflammation in a pro-inflammatory or anti-inflammatory manner. This was also observed for platelet-derived MPs (plt-EVs) [20]. In this study, at least, the levels of IL-4, IL-6, and TNF-α measured 6 h after intraperitoneal administration of PL-derived 20K-vesicles were below the measurement sensitivity, and their role in modulating inflammatory responses was not clearly demonstrated within the scope of the study.

Although the preparation and adjustment method of Plt lysate-derived vesicles was different from that in this study, Refaai et al. and Price et al. [38,39] reported on the procoagulant activity of platelet-modified lysate (PML), a plasma solution rich in human PL-derived vesicles. Their results are similar to ours, including a shortened R time (equivalent to CT in thrombo-elastometry) in thrombo-elastography and increased thrombin generation in TGA. Interestingly, they reported that the procoagulant activity of this PML was maintained even after 30 days in terms of thrombin generation, and for up to 6 months in terms of whole blood clotting. The period of storage at −80 °C for the washed platelets and platelet-derived samples used in this study was approximately two weeks. Further investigation is needed to assess functional stability after long-term storage. However, if its effectiveness remained unchanged after long-term storage, this could be a significant advantage for future clinical applications.

Our study had some limitations. A solution containing vesicle fractions was prepared through freeze–thaw cycles of the mPL solution. Generally, the methods to create platelet-derived vesicles or platelet-modulated solutions involve activation by calcium ionophores or thrombin. This study did not compare functional differences in vesicles based on the method of vesicle preparation [1]. Differences in preparation methods could potentially lead to variations in the extent of PS and Factor X expression as well as in the procoagulant activity of 20K-vesicles. Furthermore, this study did not investigate potential vascular angiogenesis and reparative effects, among other effects expected from mPL-derived vesicles. Therefore, the superiority of PL over other methods of vesicle adjustment in these aspects requires further investigation.

## 4. Materials and Methods

### 4.1. Animals

Male C57BL/6 mice (25–30 g; 12 weeks old) were purchased from Japan SLC, Inc. (Shizuoka, Japan). Mice were kept in a breeding facility with a stable room temperature (22 ± 2 °C) and a 12 h light/dark cycle (light from 07:00 a.m. to 19:00 p.m.). Food and water were provided to the mice throughout the experiment with no restrictions. Before the experiment and for at least 1 week after purchase, the mice were kept in the breeding facility to allow them to regain their strength after being transported. All mice used in this study were handled in accordance with the guidelines, and specific protocols were approved by the Institute for Disease Modeling, Kurume University School of Medicine (permission number: 2023-188).

### 4.2. Preparation of PL

mPL was prepared by first generating washed platelets and then subjecting them to repeated freeze–thaw cycles. Washed platelets were prepared as described by Aurbach et al. [40]. The procedure involved making a midline incision in the abdomen of mice anesthetized with isoflurane. The inferior vena cava was punctured with a 25-G needle, and blood was collected using a syringe containing 3% acid-citrate dextrose (1/10 volume). After adding 200 µL of PBS(-) (PBS, Cat#166-2355, FUJIFILM Wako Pure Chemical Corp., Osaka, Japan), the blood was centrifuged at 275× *g* for 5 min and allowed to stand for approximately 30 min. PRP was removed from the upper layer, and PG I2 (prostaglandin I2 sodium salt, Sigma-Aldrich, St. Louis, MO, USA) was added to the PRP immediately at a concentration of 1 μg/mL. The PRP was centrifugated at 170× *g* for 8 min, followed by centrifugation at 300× *g* for 4 min to separate and collect platelets from the PRP. The platelets were suspended in PBS containing PGI2, and the platelet suspension was centrifuged again at 300× *g* for 4 min. The collected platelets were suspended in PBS as the washed platelet solution. The upper layer without platelets, platelet-poor plasma (PPP), was used in a later step to create MpFP.

Platelets collected from one animal were suspended in 200 µL of PBS and expressed as 1 U. The concentration of washed platelets prepared using this method ranged from 120 to 143 × 10^4^/µL (measured using a Celltacα blood analyzer, NIHON KOHDEN, Tokyo, Japan) and, after being allowed to stand at room temperature for 30 min, the washed platelets were frozen at −80 °C until use. Washed platelets were prepared from three samples and used in subsequent experiments.

The mPL was prepared using the freeze/thaw cycle method wherein the washed platelet solution previously frozen at −80 °C was allowed to thaw at room temperature (22 °C) for 30 min and then refrozen at −80 °C. This process was repeated for three cycles, and the sample was centrifuged at 10,000× *g* for 1 min, followed by another centrifugation step at 2500× *g* for 15 min to remove cellular and non-thawed components, which were considered mPL.

### 4.3. Preparation of mPL-Derived Fractions

The prepared mPL was fractionated into three fractions through ultracentrifugation. Initially, a large vesicle fraction (20K-vesicles) was separated through centrifugation at 20,000× *g* for 40 min. The supernatant was centrifuged at 100,000× *g* for 90 min, and the pellets were collected as small vesicle fractions (100K-vesicles). The supernatant from this step was designated as the Plt lysate without the vesicles (PLwo-vesicles) fraction. Centrifugation at 20,000× *g* for 40 min was performed twice and the collected 20K-vesicles and 100K-vesicles were suspended in PBS to obtain suspension samples.

Similar to the washed platelets, the amount prepared from one mouse was expressed as 1 U, and the 20K-vesicles and 100K-vesicles were prepared to 40 µL/U PBS, while the PLwo-vesicles were prepared to 200 µL/U PBS. These fractions were stored at −80 °C until use.

### 4.4. Measurement of Particle Size Distribution of 20K-Vesicles and 100K-Vesicles 

The particle size distribution of the prepared vesicle fractions was measured using a Nanosight LM10 instrument (Malvern Instruments, Worcestershire, UK). Each sample of 20K-vesicles and 100K-vesicles was diluted with PBS, at 2000-fold for 20K-vesicles and 1000-fold for 100K-vesicles, and then recorded for 60 s at a flow rate of 100 µL/s. The acquired images were analyzed using Nanosight software 3.3 (Malvern Instruments, Worcestershire, UK). Each sample was measured three times.

### 4.5. TGA

#### 4.5.1. Preparation of MpFP

We evaluated the characteristics of thrombin generation in each of the three fractions derived from mPL using a commercially available fluorogenic TGA kit (Technothrombin TGA; Technoclone, Vienna, Austria). For TGA, PPP prepared in the process of preparing washed platelets was centrifuged at 10,000× *g* for 1 min, and subsequently at 2500× *g* for 15 min to remove cellular components. The supernatant was designated as platelet-free plasma (PFP). PFP was further centrifuged at 20,000× *g* for 30 min, and the supernatant was designated as MpFP. This was then diluted with PBS and adjusted to a protein concentration of 15–16 mg/mL using A280 (NanoDropC, Thermo Fisher Scientific, Waltham, MA, USA) before use.

#### 4.5.2. TGA Protocol

For the comparison of a thrombin generation capacity of 3 mPL-derived fractions, 50 µL of the TGA substrate was added to 40 µL of MpFP following the manufacturer’s instructions. As a trigger reagent for thrombin generation, each of the three fractions were added at 0.09 U/10 µL PBS. The measurements were conducted at 37 °C with continuous stirring and readings were taken every 1 min for 90 min.

To study the thrombin generation capacity of 20K-vesicles by concentration, 20K-vesicles were added to a sample containing 40 µL of MpFP and 50 µL of the TGA substrate, in steps of 0.09 U, 0.03 U, 0.01 U, and 0.003 U as the trigger reagent, and TGA was performed as above. To study the impact of PLwo-vesicles on thrombin generation, TGA was performed by adding 50 µL of the TGA substrate and 20K-vesicles (0.09, 0.06, 0.03, and 0.01 U) to 40 µL of MpFP with 0.06 U of PLwo-vesicles and 40 µL of MpFP without PLwo-vesicles, respectively.

### 4.6. Measurement of Various Platelet-Derived Proteins in the PLwo-Vesicle Fraction

The concentrations of Angiopoietin-1, DKK1, HGF, SerpinE2/protease nexin-1 (SERPINE2), and TFPI in the PLwo-vesicle samples were measured using the following ELISA kits: Mouse Angiopoietin-1 ELISA Kit PicoKine (EK1296, Boster Biological Technology, Pleasanton, CA, USA); DKK1 Mouse SimpleStep ELISA Kit (ab197746, Abcam, UK); Mouse HGF SimpleStep ELISA Kit (ab223862, Abcam, Cambridge, UK); Mouse SERPINE2 ELISA Kit (ARG82410, Arigo Biolaboratories, Hsinchu, Taiwan); and Mouse TFPI SimpleStep ELISA Kit (ab217776, Abcam, Cambridge, UK). MpFP was used as a control for comparison.

### 4.7. Characteristics of 20K-Vesicles 

The expression of PS in 20K-vesicles was analyzed using a CytoFLEX flow cytometer (Beckman Coulter, Brea, CA, USA). Gating of EVs was performed using two types of standardized beads (Megamix-Plus SSC and Magamix-Plus FSC; BioCytex, Marseille, France), and thresholds for side scatter and forward scatter were set.

A 1 µL sample of 20K-vesicles was mixed with 5 µL of Annexin V-APC (Invitrogen, 17-8007-72, Carlsbad, CA, USA) and 19 µL of the Annexin-binding buffer, making a total volume of 25 µL. The mixture was then incubated at room temperature in the dark for 15 min. After incubation, 5 µL of the sample was diluted with 1000 µL of the Annexin-binding buffer for measurement. Factor X concentration in 20K-vesicles was measured using a Factor X Total Antigen ELISA Kit (MFXKT-TOT, Molecular Innovations, Novi, MI, USA).

### 4.8. Viscoelastic Test

Viscoelastic tests continuously measure changes in blood viscosity during the blood coagulation process using whole blood as a sample [41]. To evaluate the procoagulant activity of the three mPL-derived fractions, the coagulation process upon the addition of each fraction was measured with rotational thrombo-elastometry (ROTEM Delta, Werfen, Barcelona, Spain) using citrate-added mouse whole blood. We diluted 300 µL of whole blood with PBS at a 1:2 ratio and added 20 µL of 0.2 M CaCl_2_. For the coagulation trigger, we added 0.5 U of 20K-vesicles, 100K-vesicles, or PLwo-vesicles. We evaluated the following thromboelastometric parameters: CT (time from the start of the measurement to the start of clot formation), CFT (time from the start of clot formation until an amplitude of 20 mm was reached), and MCF.

### 4.9. Tail-Snip Bleeding Assay

After the intraperitoneal administration of 20K-vesicles at 0.5 U/300 µL PBS or PBS at 300 µL, general anesthesia was induced with isoflurane 60 min later. The tail was warmed for 2 min, and then the distal 15 mm of the tail was amputated, before immediately submerging (immersing) it in 12 mL of 37 °C PBS. The time (in seconds) taken for the bleeding to stop was measured for each group of five mice. Blood loss was quantified as described by Miyazawa et al. [33]. The PBS samples containing blood were centrifuged at 500× *g* for 10 min. Afterward, 800 µL of the RBC Lysis Buffer (pluriSelect Life Science, Leipzig, Germany) was added to the pellet that had settled, and the mixture was left at room temperature (22 °C) for 10 min to ensure complete hemolysis. The absorbance was measured at 550 nm using a microplate reader (Varioskan LUX, Thermo Fisher Scientific, Waltham, MA, USA). The mice used in this study were observed for 24 h after awakening from anesthesia, and they were subcutaneously injected with 200 µL of physiological saline after confirming hemostasis.

### 4.10. Thrombin–Antithrombin Complex (TAT) and Cytokine Levels after Intraperitoneal Administration of 20K-Vesicles 

After intraperitoneal administration of 20K-vesicles at 0.5 U/300 µL PBS, a laparotomy was performed under isoflurane inhalation anesthesia either 6 h later (for IL-4, IL-6, and TNF-alpha, *n* = 5) or 24 h later (for TAT, *n* = 5), and blood was collected via a 25-G needle inserted into the inferior vena cava. The measurements of TAT and cytokines were carried out using commercially available ELISA kits as follows: TAT Complexes Mouse ELISA Kit (ab137994, Abcam, Cambridge, UK) and arigoPLEX Mouse M1/M2 Cytokines Multiplex ELISA Kit (ARG82913, Arigo Biolaboratories, Hsinchu, Taiwan). The measurements of TAT, IL-4, IL-6, and TNF-alpha were conducted following the respective kit manuals.

### 4.11. Histological Study

After conducting the tail snap-bleeding assay, the animals were observed for up to 24 h. At the 24 h time point, the animals were anesthetized again using isoflurane, and a midline incision was made in the abdomen. After confirming the absence of any abnormalities such as bleeding, the lungs, liver, and kidneys were collected and fixed in a 10% buffered formalin solution for 48 h. To evaluate the presence of microthrombi in each organ, samples were prepared using HE and PTAH staining and assessed under a microscope (OLYMPUS CX41, Tokyo, Japan).

### 4.12. Statistical Analysis

Statistical analyses were performed and figures were generated using GraphPad Prism 8.4.3 (GraphPad Software, Boston, MA, USA). For the viscoelastic test, the Friedman test was employed for the four-group comparison, followed by Dunn’s multiple comparison test for the post hoc analysis. In contrast, the Mann–Whitney U test was used for group comparisons in other experiments. Statistical significance was set at *p* < 0.05.

## 5. Conclusions

mPL-derived 20K-vesicles prepared from freeze–thawing washed platelets exhibit remarkable procoagulant activity and hemostatic effects in a tail-snip bleeding model without thrombotic complications. However, further studies are needed to investigate the hemostatic effect for more major bleeding events, such as liver injury and adverse reactions, when the dose of the 20K-vesicles is increased. In this study, only the freeze–thaw cycle method generated 20K-vesicles. For clinical use, it is necessary to investigate the best way to create 20K-vesicles with stronger procoagulant activity and methods of long-term storage. In addition, PL contains abundant bioactive biomolecules, including angiogenic growth factors, anti-inflammatory cytokines, chemokines, and antioxidants. We did not examine these advantages of the protective or regenerative effects of PL or PL-derived vesicles on injured tissue or injured endothelium in this study. Additional studies from such viewpoints may increase the utility of PL-derived vesicles in acute hemorrhagic conditions.

## Figures and Tables

**Figure 1 ijms-25-01188-f001:**
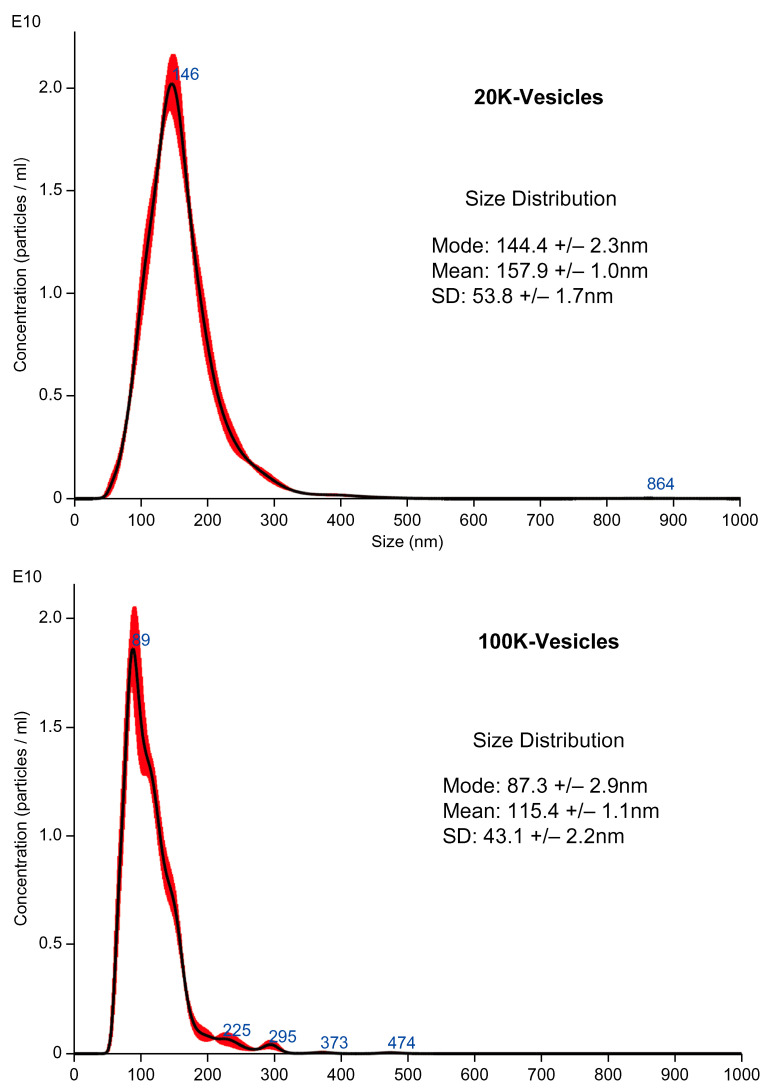
Particle distribution of the 20K-vesicles and 100K-vesicles. The most frequent diameters for the 20K-vesicles and 100K-vesicles were 144.4 nm and 87.3 nm, respectively, with clear differences between the two.

**Figure 2 ijms-25-01188-f002:**
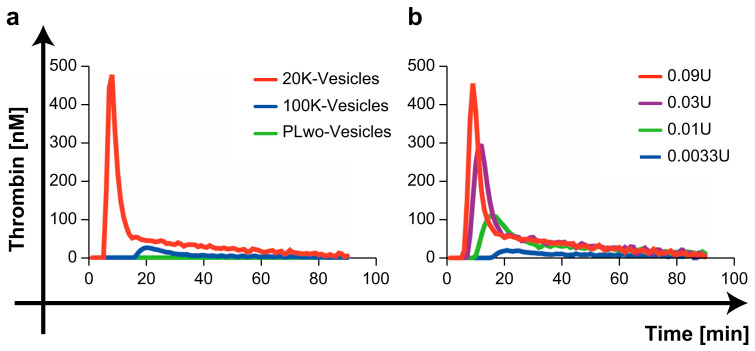
Thrombin generation assay. (**a**) The addition of the 20K-vesicles to microparticle-free plasma (MpFP) resulted in significant thrombin generation, while the addition of the PLwo-vesicle fraction did not lead to thrombin generation. A minimal level of thrombin generation was observed with the addition of the 100K-vesicle fraction. (**b**) Thrombin generation due to addition of the 20K-vesicle fraction was concentration-dependent.

**Figure 3 ijms-25-01188-f003:**
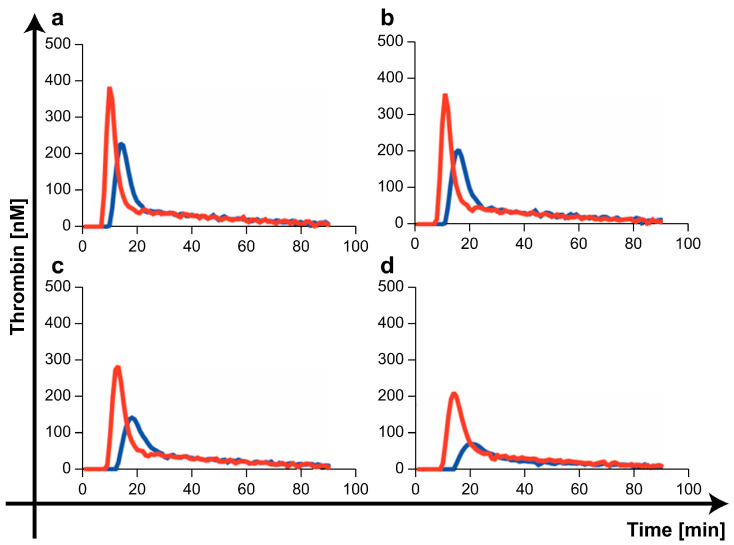
Effect of PLwo-vesicles on thrombin generation. The 20K-vesicles were added to 40 µL of MpFP with PLwo-vesicles at 0.06 U (blue) and MpFP without PLwo-vesicles (red) at various concentrations ((**a**) 0.09 U, (**b**) 0.06 U, (**c**) 0.03 U, (**d**) 0.01 U). The time to the start of thrombin generation was prolonged and the peak thrombin value was reduced when PLwo-vesicles were added to MpFP.

**Figure 4 ijms-25-01188-f004:**
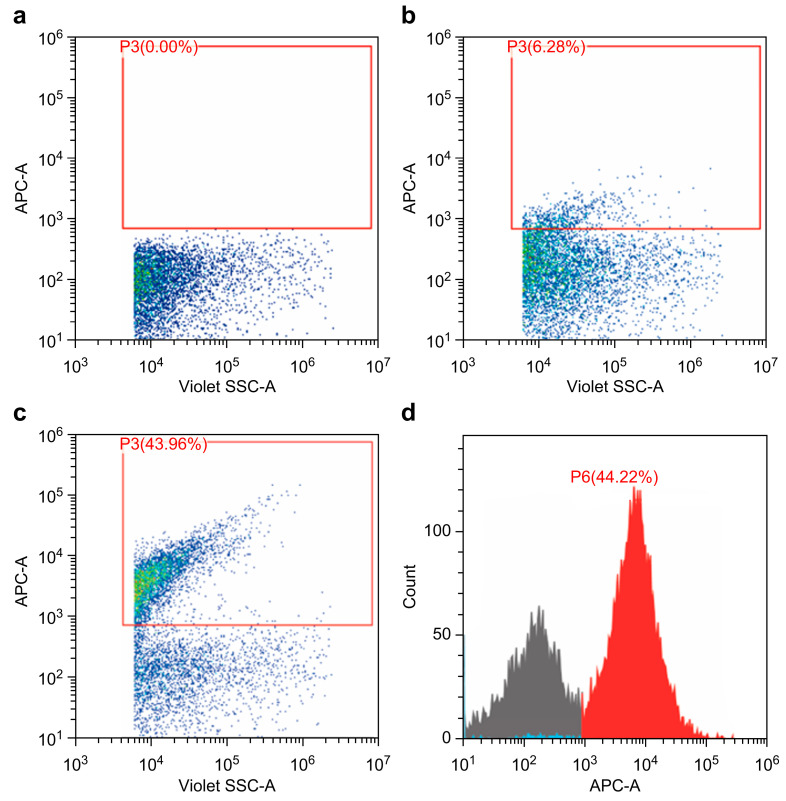
Confirmation of phosphatidylserine expression (Annexin V-positive) in 20K-vesicles. Standard beads were used to gate the distribution ranging from 100 nm to 900 nm (red frame), and each sample was analyzed at a flow rate of 10 µL/s. (**a**) The 20K-vesicles + Annexin-binding buffer. (**b**) The 20K-vesicles + Annexin V-APC (without binding buffer). (**c**) The 20K-vesicles + Annexin V-APC + Annexin-binding buffer. Annexin V positivity was confirmed in the gated vesicle regions. (**d**) Annexin V expression ratio in the 20K-vesicles (the background from the negative region includes solution noise).

**Figure 5 ijms-25-01188-f005:**
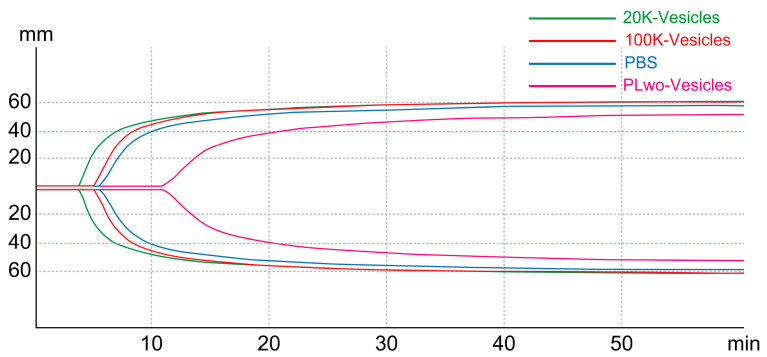
Coagulation process in ROTEM. The addition of 20K-vesicles as a coagulation trigger to mouse whole blood shortened the time to fibrin production, as represented by CT, and accelerated the rate of clot formation due to fibrin polymerization. However, the addition of PLwo-vesicles prolonged these parameters. Furthermore, the clot firmness was reduced through the addition of PLwo-vesicles.

**Figure 6 ijms-25-01188-f006:**
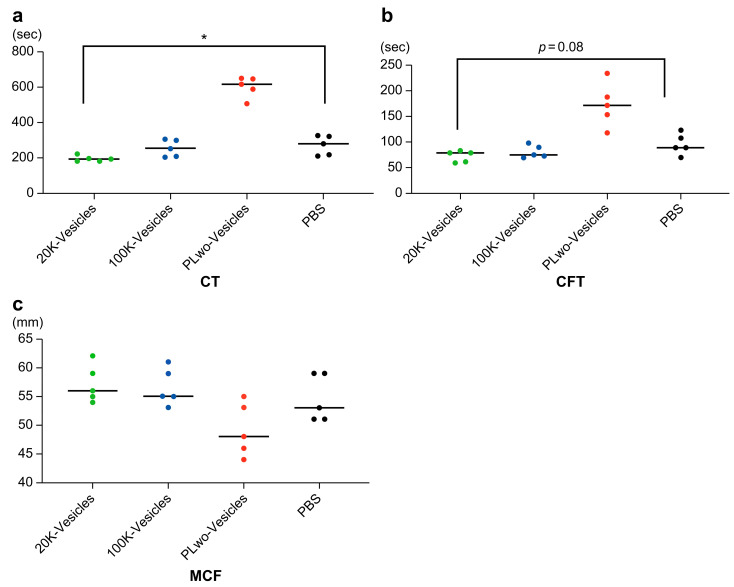
Thromboelastometric parameters in the viscoelastic test. (**a**) Clotting Time (CT), (**b**) Clot Formation Time (CFT), (**c**) Maximum Clot Firmness. Statistical analysis involved the application of the Friedman test followed by Dunn’s multiple comparison test. The Friedman test was employed to assess differences among multiple related groups, and post hoc analyses were conducted using Dunn’s multiple comparison test. * *p* < 0.05.

**Figure 7 ijms-25-01188-f007:**
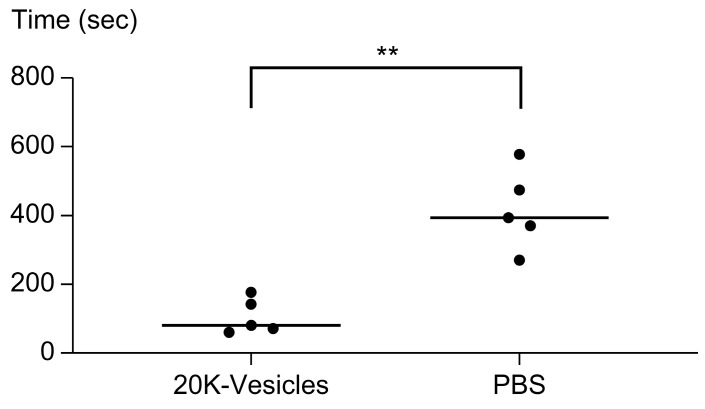
Effect of the 20K-vesicle fraction in the tail-snip bleeding assay. Intraperitoneal administration of 20K-vesicles (0.5 U/PBS at 300 µL) significantly shortened the bleeding time. ** *p* < 0.01. Statistical analysis was performed using the Mann–Whitney test.

## Data Availability

The datasets used and analyzed during the current study are available from the corresponding author on reasonable request.

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
