# Peer review of "Exploring the Hemostatic Effects of Platelet Lysate-Derived Vesicles: Insights from Mouse Models"

_ijms, 2024, doi:10.3390/ijms25021188_

Round 1
Reviewer 1 Report
Comments and Suggestions for Authors
In this manuscript, Hirayu et al compared different fractions of mouse platelet lysate and found 20K-EVs have the highest procoagulant activity whereas PLwoEVs possess anticoagulant activity. The author further showed that 20K-EVs could shorten mouse bleeding time while seemingly did not induce thrombus formation in vital organs 24hr after i.p. administration.
Comments:
(1) the order of results presentation may need to be modified. The authors may need to put “2.5 Comparison of thrombin generation capacity by mPL-derived fractions” before the earlier 2.2-2.4, then it would read more logically. (2) This journal puts the Result section before the Method section, so it is advised to define the abbreviations in the results section instead of in the method section as it is now.
(3) Although the authors define “the amount prepared from one mouse as 1 U”, this might not be informative enough. For example, how does the 20k-EV differ between different donor mice? Did the authors tested any between donor variation of the procoagulant activity of the 20K-EVs? Standardizations of this would be paramount if these EVs are to be used to stop bleeding. Maybe by EV counts, or PS-positive EVs?
(4) More details are needed for some of the experiments: how was MpFp prepared? How many number of mice or technical replicates were used per condition in the tables and figures?
(5) Line 144, the authors claimed “No apparent thrombus formation was observed in the liver, kidneys or lungs”, could the authors show these important results?
(6) The advantage of EVs over platelet concentrate is that the platelet transfusion products have a short shelf life. How does long time storage affect the procoagulant activity of 20K-EVs?
(7) Statement “This suggests the presence of particles with thrombin activity in 100K-EVs” should be changed. Thrombin-activating activity would be logic.
Author Response
Response to Review1
Comment 1,2. (1) the order of results presentation may need to be modified. The authors may need to put “2.5 Comparison of thrombin generation capacity by mPL-derived fractions” before the earlier 2.2-2.4, then it would read more logically. (2) This journal puts the Result section before the Method section, so it is advised to define the abbreviations in the results section instead of in the method section as it is now.
Response 1: Thank you for this suggestion. We agree with your comment to place Result section 2.5 (Comparison of thrombin generation ability by mPL-derived fractions) before 2.2 and have incorporated this suggestion in this document. Accordingly, we have also changed the order of some of the methods.
Response 2: Thank you for your comment regarding the definition of abbreviations. As suggested, we have changed to define the abbreviation in the Results section, where it is used for the first time.
Comment 3. Although the authors define “the amount prepared from one mouse as 1 U”, this might not be informative enough. For example, how does the 20k-EV differ between different donor mice? Did the authors tested any between donor variation of the procoagulant activity of the 20K-EVs? Standardizations of this would be paramount if these EVs are to be used to stop bleeding. Maybe by EV counts, or PS-positive EVs?
Response 3: Thank you for this suggestion. 
In this study, the dosage of vesicles, etc., was not based on the number of particles but rather on the number of units, i.e., the amount collected and prepared from the whole blood of one animal (one unit). This made it possible to compare the effects of different fractions of PL origin.
Washed platelet samples were prepared using three units per sample. Regarding the platelet-derived vesicles, the particle concentrations were measured three times in each sample, and the mean and mode values were calculated as shown in Supplement 1. We observed some between-group differences in the number of particles of the 20K-vesicles and/or 100K-vesicles in each sample; however, we considered that these did not have any major impact on the results.
In addition, we have shown the sample-specific data and the detailed measurement data on the examination of TGA and Rotem in Supplements 2 and 3.
Comment 4.  More details are needed for some of the experiments: how was MpFp prepared? How many number of mice or technical replicates were used per condition in the tables and figures?
Response 4: Thank you for your appropriate comment. Accordingly, details on how to create MpFP have been added to the Materials and Methods section. MpFP was prepared using platelet-poor plasma (PPP), which was created during the process of preparing washed platelets; hence, we obtained the same number of samples as the washed platelets. We have documented the number of mice used in each experiment in the manuscript.
Comment 5. Line 144, the authors claimed “No apparent thrombus formation was observed in the liver, kidneys or lungs”, could the authors show these important results?
Response 5: Thank you for your comment. Although no significant findings were observed, we agree with the necessity for tissue presentation.
The histological images of the kidney, lung, and liver of the 20K-vesicle administration group and the control PBS group have been provided shown in the supplement.
Comment 6. The advantage of EVs over platelet concentrate is that the platelet transfusion products have a short shelf life. How does long time storage affect the procoagulant activity of 20K-EVs?
Response 6:
As stated in the Discussion section, in this research, we conducted experiments with a maximum storage period of 2 weeks. Although it is unclear what happens to procoagulant activity during long-term storage, a previous study (reference no. 38) shows that platelet-modified lysate (a plasma solution rich in human platelet lysate-derived EVs) maintains its function even after 6 months of storage.
However, we think it will be necessary to verify once more whether the platelet lysate-derived vesicles will show similar results.
Comment 7. Statement “This suggests the presence of particles with thrombin activity in 100K-EVs” should be changed. Thrombin-activating activity would be logic.
Response 7: Thank you for your suggestion.
In the process of this research, we also conducted a preliminary study to separate two characteristically different vesicles with centrifugation at 15,000 g and 100,000 g. However, at centrifugation at 20,000 g and 100,000 g, we were more clearly able to distinguish the characteristics of large vesicles and small vesicles on the thrombin generation assay. As shown in our study, in some samples, although small vesicles separated at 100,000 g showed weak thrombin generation ability, we thought that the differences in thrombin generation ability between two different vesicles (large (20K) vesicles and small (100K) vesicles) were separated almost successfully at 20,000 g and 100,000 g.
We believe that this result shows the limitation of separating the physiological activity of vesicles by ultracentrifugation. We have briefly commented on this point in the Discussion section.

Reviewer 2 Report
Comments and Suggestions for Authors
In this manuscript Nobuhisa Hirayu and Osamu Takasu investigated effects of fractions isolated by ultracentrifugation from mouse platelet lysate on several hemostasis parameters. Whole platelet lysate are used in the clinic for treatment of various deceases. The authors analyzed different fractions of the mouse platelet lysate which were designated as 20K-EVs, 100K-EVs, and PLwoEVs according to the size of particles on several parameters of hemostasis. They showed that the 20K-EVs exhibited procoagulant activity, whereas PLwoEVs exhibited anticoagulant activity. The manuscript is clearly written and the experiments are good conducted, however there are still some questions that need additional attention.
1. The title is too ambitious. Such title the authors could have when the experiments will be done in human, not in mouse platelets.
2. Connected to 1. Why the authors used mouse platelets? They should do similar work using human platelets lysate. Even in mouse tail bleeding assay and especially for other in vitro assays will be more relevant to use human platelet lysate.
3. P. 1, L 25. “PL-derived 20K-EVs exhibited highly potent procoagulant activity, making them potential alternatives to platelets.” Not clear what the authors mean by this sentence. Making them potential alternatives for platelet transfusion, or for what?
4. P. 1, L 37-38. “Traditionally, the major function of platelets in hemostasis is to provide a membrane surface that enhances the fibrin network in blood clots and promotes coagulation.” This is not correct. It is not the major function of platelets.
5. P. 2, L. 54. “PL contains abundant EVs” This is not correct. Platelets contain different granules mitochondria, lysosomes, but they did not contain preformed abundant EVs. EVs are released for platelets, especially during their activation.
6. In the whole manuscript, the authors can not designate the fractions as EVs, they are not EVs, they are different fractions derived from platelet lysate. There are big differences between platelet lysate and EVs. And the authors should not mix them up in the whole text.
7. Results. P. 4, L 75. “2.2 Platelet intracellular proteins in PLwoEVs fraction.” There are no explanation why these proteins were chosen, from thousands of other proteins expressed in platelets, for analysis.
8. Table 1. Thromboelastometric parameters in the viscoelastic test. In the Table P values are indicated, however it is not clear which data are compared. They should be indicated by *.
9. P. 5, L. 114 -115. “The 20K-114 EVs exhibited a significant and concentration-dependent thrombin generation capacity 115 compared to the other two (Figure 4b).” For significant differences statistical data should be presented, not only original figures.
10. Discussion. P. 7, L, 155. “In the 20K-EVs, as previously reported [21–23]”. In all three references, only microparticles derived from stored platelets are described and there are not connected with the platelet lysate. Therefore, this sentence and citations are not correct.
11. P. 8, L. 207-208. “In the tail-snip bleeding assay, as in previous reports, administration of 20K-EVs significantly shortened the bleeding time.” Which previous reports, citations are need here.
12. P. 8, L. 222-224. “Additionally, no promotion of thrombus generation was observed, and there was no indication of increased thrombus formation in the liver, kidney, or lung tissues 24 h after administration compared to the control group receiving PBS.” These important data should be presented at least as Supplements.
13. 4. Materials and Methods. P. 9, L. 279. “Platelets were collected from platelet-poor plasma (PPP)” How platelets could be collected from PPP?
14. P. 9, L. 285. “120 to 143 × 104/μL” Something wrong here.
In conclusion: the authors should use for such studies human, but not mouse platelets, and they should designate the particles as lysate vesicles not EVs and do not mixed them up with real platelet derived EVs.
Author Response
Response to Review 2
Comment 1, The title is too ambitious. Such title the authors could have when the experiments will be done in human, not in mouse platelets.
Response 1: Thank you for your suggestion. We would like to change the title to “Exploring the Hemostatic Effects of Platelet Lysate-Derived Vesicles: Insights from Mouse Models.”
Comment 2.  Connected to 1. Why the authors used mouse platelets? They should do similar work using human platelets lysate. Even in mouse tail bleeding assay and especially for other in vitro assays will be more relevant to use human platelet lysate.
Response 2: Thank you for your suggestion. When considering the clinical applications for humans, and also from the standpoint of animal welfare, we believe that studies using human platelets should be considered. However, in our research, we attempted investigate the functions of platelet lysate-derived vesicles other than procoagulants not only in trauma (bleeding) models but also in mouse models of sepsis, heat stroke, and other diseases.
Ongoing studies and preliminary data have shown that the 20K-vesicles (large vesicles) that we focused on in this paper have regulatory and protective effects on vascular endothelium in some models. Although it is unclear whether similar results can be obtained with human platelets, we would like to consider further experiments using human platelet lysate in the future. Thank you for pointing this out.
Comment 3.   P. 1, L 25. “PL-derived 20K-EVs exhibited highly potent procoagulant activity, making them potential alternatives to platelets.” Not clear what the authors mean by this sentence. Making them potential alternatives for platelet transfusion, or for what?
Response 3: We appreciate the comment on this point. In accordance with the reviewer's comment, we have changed this text to “PL-derived 20K-vesicles exhibited highly potent procoagulant activity, making them potential alternatives to platelet transfusion.”
Comment 4. P. 1, L 37-38. “Traditionally, the major function of platelets in hemostasis is to provide a membrane surface that enhances the fibrin network in blood clots and promotes coagulation.” This is not correct. It is not the major function of platelets.
Response 4: Thank you for this important comment. The reviewer's comment is correct, and we apologize for the prior error. We have revised the description regarding the role of platelets in hemostasis. We have changed this text to: Hemostasis is an important physiological process that prevents bleeding after vessel injury and involves two main mechanisms—blood coagulation and platelet activation [7]. Platelets are recruited to a site of vessel injury, and the activated platelets release several proteins and molecules from α-granules and dense granules or as platelet-derived extracellular vesicles (PD-EVs), which promote continued platelet aggregation, coagulation, inflammation, and vasoconstriction. Interestingly, the surface of a type of PD-EV that is released with platelet activation has been reported to possess significantly higher procoagulant activity, approximately 50–100 times more potent than the surface of an activated platelet [8].
Comment 5. P. 2, L. 54. “PL contains abundant EVs” This is not correct. Platelets contain different granules mitochondria, lysosomes, but they did not contain preformed abundant EVs. EVs are released for platelets, especially during their activation.
Comment 6 In the whole manuscript, the authors can not designate the fractions as EVs, they are not EVs, they are different fractions derived from platelet lysate. There are big differences between platelet lysate and EVs. And the authors should not mix them up in the whole text.
Response 5 and 6: Thank you for this important comment.
We agree with you and have accordingly incorporated this suggestion throughout our paper.
In the initial draft, we described vesicles isolated from platelet lysate as "extracellular vesicles" without deep thought even though they did not release spontaneously or did not have physiologically activated processes. 
Contrarily, as the expression of PS and lipid membrane components (data not shown) were confirmed on the vesicle-like products described as 20K EVs in the first draft, we think that they should be described as PL-derived vesicles.
In accordance with the original definition of extracellular vesicles, we have corrected the term to "mPL-derived vesicles" throughout the new draft.
In addition to this, we have cited a new reference (no. 22).
We would appreciate it if you could review the corrected manuscript again.
Comment 7 Results. P. 4, L 75. “2.2 Platelet intracellular proteins in PLwoEVs fraction.” There are no explanation why these proteins were chosen, from thousands of other proteins expressed in platelets, for analysis.
Response 7.: We agree that the reason why we selected proteins such as Angiopoietin-1, HGF, and DKK-1 requires clarification. To determine the extent to which bioactive proteins present in α-granules, including SerpinE2/protease nexin-1, are released during the platelet lysate preparation process, we measured the concentrations of Angiopoietin-1, HGF, and DKK-1, which can be stably measured using commercially available kits, and compared them with those in plasma (microparticulate free plasma, MpFP). We selected these proteins as they are likely to be involved in hemorrhage or posthemorrhagic pathology, including vascular regulation, regeneration, leukocyte infiltration, but are not directly relevant to this study.
We have added the following text to the Discussion (page 9, lines 264–270): Considering the isolation process used in this study, we hypothesized that various physiologically active substances present in platelet granules would be found at high concentrations in PLwo-vesicles. Therefore, the concentrations of TFPI and serpinE2/protease nexin-1 in PLwo-vesicles, along with representative proteins known to be present in É‘ granules, such as Angiopoietin-1, HGF, and Dickkopf-1 [DKK 1], which can be stably measured using commercially available kits, were examined, with MpFP serving as a control.
Comment 8. Table 1. Thromboelastometric parameters in the viscoelastic test. In the Table P values are indicated, however it is not clear which data are compared. They should be indicated by *.
Response 8. Thank you for your comment. Regarding the thromboelastometric parameters, Table 1 has been changed to a figure (Figures 5 and 6 and Supplement 3) to more clearly show differences between the groups.
Regarding clotting time (CT), significant differences among the four groups (20K-vesicles, 100K-vesicles, PLwo vesicles, and PBS [control]) were confirmed during analysis using the Friedman test. Dunn's multiple comparison test was performed as a post hoc analysis (multiple comparisons), and a statistically significant difference was observed between the 20K-vesicles and PBS groups. In other words, CT was significantly shorter for the three groups compared with for the control group. After the Friedman test, when comparing each parameter using Dunn's multiple comparison test, as illustrated in Supplement 2, significant reductions were observed in CT and CFT groups between the 20K-vesicles and 100K-vesicles compared with in the PLwo-vesicles. In the MCF group, there was a significant decrease in the PLwo-vesicles compared with in the 20K-vesicles and 100K-vesicles.
Comment 9. P. 5, L. 114 -115. “The 20K-114 EVs exhibited a significant and concentration-dependent thrombin generation capacity 115 compared to the other two (Figure 4b).” For significant differences statistical data should be presented, not only original figures.
Response 9: Of the three PL-derived fractions, 20K-V, 100K-V, and PLwo-vesicles, only t20K-vesicles showed thrombin generation ability on TGA. We provided a table to show detailed data of the thrombin generation assay in the Supplement. A significant difference among the three groups is confirmed in the statistical analysis of variance test; however, a post hoc comparison (multiple comparison test) could not show a statistically significant difference. We believe that by increasing the number of experiments, it is possible to show a clear difference statistically; nevertheless, we think that the difference is sufficiently convincing from the original graph and additional supplement data.
Comment 10. Discussion. P. 7, L, 155. “In the 20K-EVs, as previously reported [21–23]”. In all three references, only microparticles derived from stored platelets are described and there are not connected with the platelet lysate. Therefore, this sentence and citations are not correct.
Response 10: Thank you for your comment. As you pointed out, the vesicles in the cited references [23-25, new No] are microparticles derived from stored platelets, which are completely different from the large vesicles (20K-vesicles) derived from platelet lysate, which we focused on. Therefore, we have changed the text in the Discussion to: Although the 20K-vesicles in this study are vesicles that have not undergone an in vivo physiological activation, significant procoagulant activity was observed in them. Moreover, this physiological activity was also similar to the platelet transfusion product-derived vesicles [23-25].
Comment 11. P. 8, L. 207-208. “In the tail-snip bleeding assay, as in previous reports, administration of 20K-EVs significantly shortened the bleeding time.” Which previous reports, citations are need here.
Response: Thank you for your appropriate comments. We agree with the relevance of this reference and have added it to the Discussion and reference [33]. Vesicles that were used in it are precisely different from the 20K-vesicles we administered; therefore, we have also discussed this perspective.
We have changed the text to: In the tail-snip bleeding assay, administration of 20K-vesicles significantly shortened the bleeding time. In the 20K-vesicles group, the sustained oozing bleeding time, which was observed in the late phase of bleeding in the PBS treatment group, was significantly shorter. We assume that this may be the reason for the shortened bleeding time. Contrarily, a previous report [33] that examined the function of platelet-derived vesicles collected from stored human platelet blood in a similar tail snip bleeding assay reported that the administration of platelet-derived vesicles reduced the amount of bleeding. In this study, a reduction in blood loss could not be demonstrated. We amputated the tail more proximally, at a length of 15 mm, compared with previous studies [33]. This resulted in the resection of a larger diameter blood vessel, leading to significant amount of bleeding immediately after vascular resection (early phase of bleeding), which then strongly influenced the total bleeding. We considered this as the reason why no significant difference was observed in the amount of total bleeding between the two groups. Further research is required to determine whether the difference in blood loss compared with those in previous studies is due to differences in the coagulant activity of platelet-derived vesicles.
Comment 12. P. 8, L. 222-224. “Additionally, no promotion of thrombus generation was observed, and there was no indication of increased thrombus formation in the liver, kidney, or lung tissues 24 h after administration compared to the control group receiving PBS.” These important data should be presented at least as Supplements.
Response 12: Thank you for your comment. Histological images of the kidney (x200), lung (x100), and liver (x200) have been provided in the Supplement. Although all tissue samples that were stained with HE or TPHA were evaluated blindly, no obvious thrombus or microthrombi formation was observed in any tissue.
Comment 13. 4. Materials and Methods. P. 9, L. 279. “Platelets were collected from platelet-poor plasma (PPP)” How platelets could be collected from PPP?
Response 13: As per the reviewer's comment, we agree that the expression was confusing, and we apologize for this issue. Hence, we have now revised the text. Since platelets were collected by centrifuging PRP in two steps, the text has been revised as follows: PRP was removed from the upper layer, and PG I2 (prostaglandin I2 sodium salt, Sigma-Aldrich, STL, USA) was added to the PRP immediately at a concentration of 1 μg/mL. The PRP was centrifugated at 170 g for 8 min, followed by centrifugation at 300 g for 4 min to separate and collect platelets from the PRP. The platelets were suspended in PBS containing PGI2, and the platelet suspension was centrifuged again at 300 g for 4 min. The collected platelets were suspended in PBS as the washed platelet solution. The upper layer without platelets, platelet-poor plasma (PPP), was used in a later step to create MpFP.
Comment 14. P. 9, L. 285. “120 to 143 × 104/μL” Something wrong here.
Response 14: As you pointed out, there is a mistake; we apologize for this issue. We have corrected “ranged from 120 to 143 × 104/µL” to "ranged from 120 to 143 × 104/µL."

Round 2
Reviewer 1 Report
Comments and Suggestions for Authors
The authors have adequately addressed my previous comments, i have no more comments.
Reviewer 2 Report
Comments and Suggestions for Authors
The authors corrected manuscript according to the comments, and I have no more questions.